# Socio-Economic and Environmental Determinants of Malnutrition in under Three Children: Evidence from PDHS-2018

**DOI:** 10.3390/children9030361

**Published:** 2022-03-04

**Authors:** Muhammad Shahid, Yang Cao, Muhammad Shahzad, Rafit Saheed, Umara Rauf, Madeeha Gohar Qureshi, Abdullah Hasnat, Asma Bibi, Farooq Ahmed

**Affiliations:** 1School of Insurance and Economics, University of International Business and Economics (UIBE), Beijing 100029, China; de202159006@uibe.edu.cn; 2Department of Anthropology, Bahauddin Zakariya University, Multan 60800, Pakistan; mshahzad@bzu.edu.pk; 3School of Public Policy, Pakistan Institute of Development Economics, Islamabad 44000, Pakistan; rafitsaheed.19@pide.edu.pk; 4Department of Psychology, GC Women University, Sialkot 2021, Pakistan; umara.rauf@gcwus.edu.pk; 5Department of Economics, Pakistan Institute of Development Economics, Islamabad 44000, Pakistan; madeeha.qureshi@pide.org.pk; 6Medical and Dental College, Bahria University, Karachi 74400, Pakistan; abdullahhasanat3@gmail.com; 7Independent Researcher in Applied Psychology, Lahore 54000, Pakistan; asmatoorkham@gmail.com; 8Department of Anthropology, Quaid-i-Azam University, Islamabad 44000, Pakistan; jam007@uw.edu; 9Department of Anthropology, University of Washington, Seattle, WA 98195, USA

**Keywords:** malnutrition, sanitation facility, water source, wealth quintile index, interaction terms, Pakistan

## Abstract

Objectives: This research investigates the association of malnutrition with social and economic factors in general and environmental factors in specific such as sanitation facilities and drinking water sources for Pakistan. Methods: Authors used the latest data of 1010 Under-Three children from Pakistan Demographic and Health Survey (PDHS) 2017–2018. Cumulative Index of Anthropometric Failure (CIAF) was developed to measure the malnutrition status among children based on z-scores of WHZ, WAZ, and HAZ, respectively. The study has applied the discrete-choice logistic methodology to find the relationship of malnutrition with socio-economic characteristics. The interaction terms of drinking water source and sanitation facility have been measured to see the impact of environmental factors on child malnutrition. Results: The study results depict that the likelihood of malnutrition increases when the child had diarrhea recently and the child belongs to the deprived region such as KPK, Sind, and Baluchistan. However, the chances of child malnutrition drop with (1) an escalation of mothers’ education, (2) a rise in wealth status of the household, and (3) the improved water source and sanitation facility in the household. The only water-improved sanitation category of the interaction term is significant in the model which depicts that households having both improved water and improved sanitation facilities had very fewer chances of malnutrition among their children. Conclusion: Authors conclude that malnutrition in younger children is associated with improved water as well as sanitation facilities, maternal education, and household wealth in Pakistan.

## 1. Introduction

Malnutrition is a multi-dimensional problem that is caused by multiple socio-economic and environmental determinants. The prevalence of malnutrition is highest in low- and middle-income countries [1]. In the South Asian region, the stunting prevalence is highest with 26.9 million (38.9%) of the global wasted children [2]. Malnutrition prevalence in India, Bangladesh, and Pakistan has been over the threshold limit of 15% wasting (weight for height), 30% stunting (height for age), and 10% underweight (weight for age) [3,4]. In Pakistan, according to the PDHS 2017–2018 report, 38% of children are stunted, 23% are underweight, and 8% are wasted.

Malnutrition impacts the development of children in many ways which varies based on socio-economic background and type of malnutrition [5]. For Example, the earnings by stunted children have been estimated at below 20 percent than those of non-stunted counterparts, and their likelihood to live in poverty has been above 30 percent [6]. The economic loss of malnutrition is very high in developing countries. Every year poverty and malnutrition cost countries huge economic loss in terms of loss in GDP [7]. In Asian and African regions nearly 11% of loss in GDP is due to different forms of malnutrition such as stunting and being underweight [8]. The economic returns of investing in nutritional programs are very high. Investing USD 1 in stunting reduction intervention in developing countries, an economic return of USD 30 is expected [9].

Some studies on child malnutrition found that water, sanitation, and hygiene (WASH) is a prominent determinant in South Asia [10,11,12], including Pakistan [13,14]. Children and infants are more prone to malnourishment and stunting because of frequent exposure to diarrhea and other gastrointestinal infections, which are associated with open defecation and bad WASH conditions [15,16]. Studies show that when WASH conditions are insufficient, children cannot perform well or remain absent from schools, and the chances of sexual assault and rape amplify for girls [17,18,19]. The WHO and UNICEF support improved sanitation, water, and hygiene as a primal strategy for battling child morbidity, mortality, and diarrhea, especially in developing countries [20]. Children with moderate or high wasting are at a greater risk of mortality [2]. SDG goal 3 ensues a reduction in child mortality. Pakistan could not achieve the target set for MDGs to reduce child mortality and still struggling to meet and achieve the targets of child mortality set by SGDs [21].

This research is essential on two accounts. First, there is a dearth of evidence on determinants of malnutrition in Under-Three children focusing in specific on socioeconomic and environmental factors utilizing the latest 2018 PDHS data set. Second, this study measures the interaction of sanitation and water facility on child malnutrition. Therefore, the research hypothesis is that malnutrition in Under-Three children is associated with many proximate factors, especially environmental indicators. Literature is available on water sources and sanitation facilities in Pakistan while no study used interaction terms to analyze the combined effects of sanitation and water facility in the case of Pakistan, which is the gap this study has covered. Above all the results of the study may be used in identifying the public policy solutions for combating child malnutrition and morbidity.

## 2. Study Materials and Methods

### 2.1. Theoretical Framework

Most of the researches on the assessment of child malnutrition pursue the utility-maximizing model by postulating the production function of households [22,23]. This model assumes that each child lives in a unit called a household.

*N^i^* = n [*H*, *Z*, *W*, *C*, ε]

*N^i^* is taken as the standard measurement of anthropometry for a child. *C* shows consumption, *W* represents the vector of specific children and maternal factors; *H* represents the vector of specific household and environmental factors; *Z* represents the vector of health factors, while the error term of children-specific is ε.

The reduced specified form of the nutrition production function for the study is:

CIAFi = f (household factors and environmental factors, disease factors, child and maternal specific factors, socioeconomic factors, individual factors, εCIAF).

### 2.2. Data-Set and Description

For this study, PDHS 2017 to 2018 data set was used for the analysis with a sample of 1010 children aged less than three years. This data provided wide information on nutrition and demographic characteristics, women, and children nutritional and healthcare information, women empowerment, domestic violence, etc. The study used anthropometric measurements of eligible Under-Three children. The statistical analysis was conducted using anthropometric measurements. Furthermore, data on household characteristics, child and mother specific factors, child disease factors, and environmental and socioeconomic characteristics were used in this study.

### 2.3. Outcome Variable

The study used information on the children’s height, weight, and age from the PDHS 2017–2018 to construct indices. As per WHO (2009) standards for growth [24], three indices are used to measure child malnutrition. (i) Stunting (HAZ), is defined as “if z-scores of height for age is <−2 S.D. of the median value of World Health Organization (WHO) guidelines”; (ii) Underweight (WAZ), “if z-scores of weight for ages is <−2 S.D. of the median value of World Health Organization (WHO) standards;”, (iii) Wasting “if z-scores of weight for heights is <−2 S.D. of the median value of World Health Organization (WHO) guidelines”.

In this study, to measure child malnutrition, the CIAF was developed. The CIAF classification divides the children into seven (7) different categories: (1) No Failure, (2) Only stunted, (3) Only wasted, (4) Only under-weight, (5) Both (stunted and underweight), (6) Both (wasting and underweight), and last is (7) (stunting, wasting, and underweight). The total measure of child malnutrition prevalence is calculated by combinations of all except group A. The binary variable “1” is used if a child is malnourished otherwise use “0” if a child is not malnourished.

### 2.4. Proximate Determinants of Malnutrition

The explanatory variables of this study are defined and reported in Table 1.

### 2.5. Statistical Analysis

The study developed a hypothesis that malnutrition in younger children depends on many proximate indicators (among them, especially water and sanitation). Logistic regression was used to measure the relationship between proximate variables (socioeconomic and environmental variables) and the outcome variable (child’s malnutrition). The study develops the CIAF index to measure child nutrition status.

The explanation of the binary logistic model is:CIAFin = Yin = [1 = if the child is malnourished, and 0 = otherwise]

In binary form CIAFin and Yin are equal. The response is either success or failure more easily either child malnourished or not which is coded 0 or 1 value. However, here in this study Y is child malnutrition status and study assessing its association with multiple explanatory variables (x1, x2,…, xn). The model specification and reduced form of multivariate logistic regression which estimates the probability scores and odds ratios of the outcome variable (CIAF) conditioned on explanatory variables is:P (Yi = 1|X1i, X2i,…, Xkn) = F (β0 + β1 X1i + β2 X2i +…+ βn Xkn)

In the above equation, Yi denotes the outcome variable (CIAF), Xi represents the explanatory variables, coefficients are β’s, which explains the degree of association with the dependent variable CIAF, while the error term is ε.

Three levels of significance were used in the logistic regression to show the significant relationship of the independent variable with the outcome variable: *p* < 1%, *p* < 5%, and <10%. The empirical testing has been performed on Stata version 15.

## 3. Results

### 3.1. Descriptive Statistics

This research examined the impact of sanitation and water quality on child malnutrition. PDHS data were analyzed for each variable. The percentage of CIAF occurrence in a child was given according to different characteristics in Table 2, which showed that the prevalence of malnutrition was almost the same in both sexes. However, higher malnutrition was observed in the third year of their childhood. Nutrition vulnerability in Sindh and Balochistan is higher than in other provinces. Mother employment and her decision in spending results in decreasing the malnutrition vulnerability. Additionally, literacy has a huge effect on the nutritional status of children. Mothers without education had the highest percentage of malnourished children. Poverty also determines the nutrition of a child. Two-thirds of the total malnourishment among children prevailed in poor families.

The data analysis indicated that undernutrition prevalence was (26.45%) among children who had an un-improved source of water in their houses. Similarly, malnutrition was (36.27%) among children who had un-improved sanitation facilities in their houses.

### 3.2. Association of Malnutrition with Water Source and Sanitation Facility

Figure 1 highlights the stunting, wasting, and underweight prevalence by type of sanitation facility. Results in Figure 1 show that stunting and underweight rates decrease as households have improved sanitation facilities but wasting rates remain constant over the change in sanitation facilities.

Figure 2 shows the stunting, underweight, wasting prevalence by water quality. Results in Figure 2 show that stunting and underweight rates decrease as households have improved source of water but wasting rates remain constant over the change in the source of water.

In Table 3, the result estimations of logistic regression have been reported. The logistic regression estimated the covariates of the CIAF for the overall sample. In the results, the age of children from 19–24 months was linked with the high number of malnutrition odds (CIAF) in the case of preschool children (OR = 2.51, 95% CI: 1.54–4.12). Similarly, the chances of becoming malnourished were also higher for the 25–36 months’ older child group (OR = 2.86, 95% of CI: 1.93–4.26). The region Sindh was connected with a high number of odds of malnutrition (CIAF) among preschool children (OR = 2.82, 95% CI: 1.72–4.60), chances of becoming malnourished among under-three children were also higher in KPK (OR = 1.76, 95% CI: 1.06–2.91) and also higher in Baluchistan region (OR = 3.02, 95% CI: 1.79–5.06). For mothers who had attained primary (OR = 0.66, 95% CI: 0.42–1.04) and high school (OR = 0.40, 95% of CI: 0.23–0.72), associated with a low number of odds of malnourishment than other educational categories. The malnutrition odds were supposed to be high for preschool children who had diarrhea recently (OR = 1.48, 95% CI: 1.07–2.04). Across the wealth quantiles, the malnutrition odds were low in the higher WIQ (OR = 0.58, 95% of CI: 0.34–0.99), and least in the highest WIQ (OR = 0.40, 95% CI: 0.22–0.74). The households with improved water sources had OR = 0.71 with 95% CI: 0.51–1.01 leads to a lesser prevalence of malnutrition among their children. Likewise, the households with improved sanitation facilities had an OR = 0.69 with 95% CI: 0.52–0.91 suggests low chances of malnutrition.

The results of interaction terms for water and sanitation depict that only improved water and improved sanitation categories are significantly contributing to malnutrition reduction in children. It shows that households who have both improved water and sanitation had less likelihood to encounter the prevalence of malnutrition among their children (OR = 0.65, 95% CI: 0.42, 1.02).

## 4. Discussion

The empirical estimations illustrated that mothers with primary and high education had lesser chances of malnutrition among their children. Some of the previous studies also support the results that mothers’ education is a significant determinant of malnutrition [5,25,26,27]. The results showed that mothers having their body mass greater than equal to 18.5 kg/m^2^, had (OR = 0.50, 95% CI: 0.25, 1.02) lower odds of malnutrition among their children. These results are compatible with some studies conducted across the world [28,29,30]. Additionally, results showed that the children of age groups 19–24 months and 25–36 months had a higher probability of being malnourished as compared to other age groups. It can be interpreted as when the age increases the prevalence of malnutrition also increases [30,31].

Diarrhea, one of the major child malnutrition and infant mortality, becomes more dangerous if the child belongs to a deprived family [32]. The study indicated that children who had diarrhea recently increased the probability of malnutrition. The finding is aligned with some studies which show that children who had diarrhea recently had a higher probability of malnutrition prevalence [28,33]. The household wealth index quintile has been used as a wealth status indicator. In the study, the result estimations showed a strong and significant association between malnutrition and wealth status. The results revealed a declining trend in malnutrition prevalence rates across households in different wealth indices. Malnutrition prevalence was the lowest among richer and richest households [34,35,36,37]. The studies correlate with the analysis that in comparison to poor households, children are at a lower prospect of becoming malnourished when they possess more resources to purchase food items [38,39,40].

The results for the regions showed that children belonging to Sindh, KPK, and Baluchistan provinces had higher chances of becoming malnourished. Variation in nutrition existed on a regional basis. One study shows that in Pakistan, higher malnutrition exists among females in Baluchistan and Sindh compared to KPK and Punjab [41]. Another study depicted that all coexisting forms of malnutrition in Sindh and Baluchistan are higher compared to the other regions of Pakistan [3]. The reason is that the Baluchistan and Sindh provinces of the country remained neglected in terms of human development in the past.

Importantly, the study showed that when the household had improved water along with improved sanitation facilities, the chances of malnutrition among children decreased. Empirical evidence from the African and Asian regions shows that hygienic water for drinking and improved and quality sanitation facilities significantly reduce the degree of morbidity, mortality, and malnutrition among younger children [10]. A study in Cameroon (central Africa) depicted that, the likelihoods of diarrhea were higher among the children and toddlers of those households who have limited access to water and had improper sanitation facilities [42]. Using 2005 to 2009 DHS data sets a study for 11 countries of sub-Saharan Africa depicts that water and sanitation facilities’ combined impact is more likely to reduce the chances of malnutrition than the water and sanitation facilities’ separate impact [43]. Another research in Ghana, West Africa depicts that improved sanitation, water, and hygiene reduce the risk of stunting by 15 percent [44]. Additionally, a study in Indonesia highlights that the likelihoods of stunting are three times higher if households drink un-improved water and have un-improved toilets [45]. A study from India highlights that children are 2% less likely to achieve a minimum standard of dietary diversity who live in poor water and sanitation households [46]. In India, poor access to water and sanitation is associated with lower birth weight in infants and preterm birth [47]. Evidence shows that WASH might impact child development, principally via stunting [12].

Few of the regional studies in Pakistan highlight that improved water and sanitation are significant predictors of malnutrition [48,49,50] and the incidence of diarrhea and malnutrition in children is quite high for those households with unhygienic drinking water and unimproved sanitation facilities [51]. Studies highlighted that Pakistan needs to invest in providing water and sanitation [52]. Water and sanitation are considerably linked with malnutrition in Pakistan and the progress in water and sanitation is low in the SAARC region [53].

### Limitations

This study has examined only the socio-economic and environmental determinants of malnutrition among under-three children in Pakistan. However, child malnutrition can also be influenced by several other significant factors which need to be explored in future studies. Cultural and structural variables can also play important role in determining malnutrition among under-three children in Pakistan.

## 5. Conclusions

This study suggests that malnutrition in younger children is associated with multiple socioeconomic and environmental determinants in Pakistan. The results of this research show that significant reasons for under-three malnutrition in Pakistan are regional disparities, diarrhea, maternal education, wealth index, unimproved water source, and unimproved sanitation facility in the households. Therefore, the study urges a redirection of resources from megaprojects towards community development projects such as investment in improving water and sanitation facilities, as well as education and employment for women.

## Figures and Tables

**Figure 1 children-09-00361-f001:**
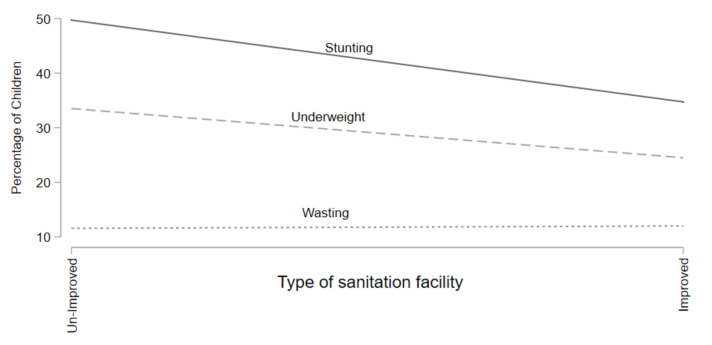
Prevalence of Stunting, Underweight, & Wasting for Sanitation Facility.

**Figure 2 children-09-00361-f002:**
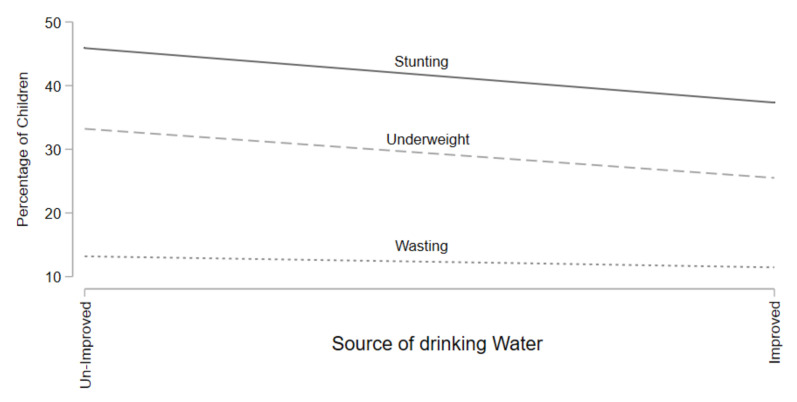
Prevalence of Stunting, Underweight, and Wasting for Water Source.

**Table 1 children-09-00361-t001:** The definitions and the descriptions of different indicators.

Variable	Definition	Mean	SD
CIAF	=1 if child is malnourished, =0 otherwise	0.4941	0.5002
Sex of child	=1 if male, =2 if female	1.500	0.5008
Age of child	=1 if 0–6 months; =2 if 7–12 months; =3 if 13–18 months; =4 if 19–24 months; =5 if 25–36 months	3.394	1.560
Birth order number of child	Measured as a continuous variable	2.531	1.193
Mother employment status (in the past 12 months)	=1 if employed, =0 if unemployed	0.115	0.319
Mother BMI	=0 if BMI < 18.5 kg/m^2^, =1 otherwise	0.891	0.312
Decision to spend women own earning	=1 if yes, =0 otherwise	0.096	0.295
Mother education	=0 if no education; =1 if primary; =2 if middle; =3 if higher	0.7439	1.095
Residence	=1 if Urban, =2 if Rural	1.599	0.4899
Provinces/Region	=1 if Province Punjab, =2 if Province Sindh, =3 if Province KPK, =4 if Province Baluchistan, =5 if ICT, =6 if AJK, =7 if erstwhile FATA	3.899	2.258
HH Size	=1 if less than 5 mem, =2 if 6–10 mem, =3 if 11–15 mem, =4 if greater than 15 mem	2.431	0.8334
Wealth quintile index	=1 if poorest; =2 if poor; =3 if middle; =4 if richer; =5 if richest	2.517	1.406
Episode of diarrhea	=1 if yes, 0 otherwise	0.1998	0.3999
Source of drinking water	=1 if improved, 0 otherwise	0.758	0.428
Type of Sanitation	=1 if improved, 0 otherwise	0.681	0.466

Mem = Members.

**Table 2 children-09-00361-t002:** Descriptive analysis describing the association between different socioeconomic characteristics over CIAF (child malnutrition).

Variables	Categories	Frequencies	Percentages
Sex of child	Male	252	50.50
	Female	247	49.50
Age of Child (Months)	0 to 6	69	13.83
	7 to 12	60	12.02
	13 to18	62	12.42
	19 to 24	81	16.23
	25 to 36	227	45.49
Birth Order Number	Birth order 1	99	19.84
	2 or 3	167	33.47
	4 or 5	120	24.05
	6 or 7	69	13.83
	Above 7	44	8.82
The decision to Spend Women Earning	Not-Involved	441	88.38
	Involved	58	11.62
Region	Punjab	49	9.82
	Sindh	123	24.65
	KPK	77	15.43
	Baluchistan	124	24.85
	Gilgit Baltistan	23	4.61
	ICT (Capital)	14	2.81
	AJK	26	5.21
	FATA	63	12.63
Mother Body Mass Index	Less than 18.5 kg/m^2^	64	12.93
	≥18.5 kg/m^2^	431	87.07
Maternal Education Level	Illiterate	365	73.15
	Primary	48	9.62
	Middle	59	11.82
	High	27	5.41
Working Status of Mothers	Not-Employed	423	84.77
	Employed	76	15.23
Size of Household	Less than 5 members	41	8.22
	6–10	225	51.10
	11–15	130	26.05
	Greater than 15	73	14.63
Wealth Index	Poorest	189	37.88
	Poorer	146	29.26
	Middle	74	14.83
	Richer	52	10.42
	Richest	38	7.62
Source of Drinking Water	Un-Improved	132	26.45
	Improved	367	73.55
Had Diarrhea Recently	No	368	73.75
	Yes	131	26.25
Type of Sanitation Facility	Un-Improved	181	36.27
	Improved	318	63.73

Source: Author’s estimation.

**Table 3 children-09-00361-t003:** Adjusted odds ratios for covariates of the CIAF based on logistic regression.

Variables	Categories	Coef.	Odds Ratio	Std. Err.	95% CI
Sex of Child	Male (R)				
	Female	−0.098	0.9065	0.1268	(0.689, 1.19)
Age of Child (in months)	0 to 6 months (R)				
	7–12 months	0.176	1.192	0.295	(0.733, 1.94)
	13–18 months	0.156	1.169	0.286	(0.724, 1.89)
	19–24 months	0.922	2.514 ***	0.634	(1.54, 4.12)
	25–36 months	1.05	2.863 ***	0.580	(1.93, 4.26)
Birth Order Number	Measure as continuous	−0.02	0.9798	0.171	(0.696, 1.38)
Decision to Spend Women Earning	Not-Involved (R)				
	Involved	0.265	1.304	0.483	(0.631, 2.69)
Region	Punjab (R)				
	Sindh	1.035	2.816 ***	0.705	(1.72, 4.60)
	KPK	0.564	1.758 ***	0.453	(1.06, 2.91)
	Balochistan	1.103	3.012 ***	0.798	(1.79, 5.06)
	Gilgit Baltistan	−0.138	0.871	0.304	(0.439, 1.72)
	ICT (Capital)	0.0636	1.066	0.424	(0.489, 2.32)
	AJK	0.1358	1.145	0.38	(0.598, 2.19)
	FATA	0.1518	1.164	0.337	(0.660, 2.05)
Mother’s Education Level	Illiterate (R)				
	Primary	−0.421	0.656 **	0.15	(0.415, 1.04)
	Middle	−0.326	0.72	0.167	(0.459, 1.14)
	High	−0.907	0.40 ***	0.118	(0.227, 0.72)
Mother’s Working Status	Not-Employed (R)				
	Employed	−0.021	0.9791	0.326	(0.509, 1.88)
Mother Body Mass Index	Less than 18.5 kg/m^2^ (R)				
	≥18.5 kg/m^2^	−0.333	0.717	0.167	(0.454, 1.13)
Household Size	<5 members (R)				
	6–10	0.154	1.167	0.308	(0.695, 1.96)
	11–15	0.280	1.323	0.379	(0.754, 2.32)
	Greater than 15	0.156	1.169	0.361	(0.638, 2.14)
Wealth Index	Poorest (R)				
	Poorer	−0.118	0.889	0.178	(0.599, 1.32)
	Middle	−0.293	0.746	0.186	(0.458, 1.22)
	Richer	−0.545	0.579 **	0.159	(0.338, 0.995)
	Richest	−0.91	0.403 ***	0.125	(0.219, 0.738)
Had Diarrhea Recently	No (R)				
	Yes	0.389	1.475 ***	0.244	(1.07, 2.04)
Source of Drinking Water	Un-Improved (R)				
	Improved	−0.335	0.715 **	73.55	(0.505, 1.01)
Type of Sanitation Facility	Un-Improved (R)				
	Improved	−0.372	0.689 ***	0.098	(0.521, 0.912)
Water # Sanitation (interaction)	Un-Improved Water # Un-Improved Sanitation (R)				
	Un-Impr-water # Impr-sanit	−0.068	0.934	0.238	(0.57, 1.54)
	Impr-water # Un-Impr-sanit	0.203	1.23	0.276	(0.79, 1.90)
	Impr-water # Impr-sanit	−0.308	0.735 **	0.137	(0.51, 1.06)
**Model Overall Significance**					
Number of observations = 1010		Prob > Chi^2^ = 0.0000			
LR-Chi^2^(29) = 172.22		Pseudo-R^2^ = 0.1238			
References: Odd ratios; Confidence Intervals					
(R): shows the reference category					
Level of Significance: *** if *p* less than 0.01 ** if *p* less than 0.05					

Source: Authors’ estimations.

## Data Availability

This study utilized the secondary data of the Pakistan Demographic and Health Survey 2017–2018. Available online https://dhsprogram.com/data/dataset/Pakistan_Standard-DHS_2017.cfm?flag=1 (accessed on 15 June 2021).

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
