# Peer review of "Socio-Economic and Environmental Determinants of Malnutrition in under Three Children: Evidence from PDHS-2018"

_children, 2022, doi:10.3390/children9030361_

Round 1

Reviewer 1 Report

General statement:

The study addresses the association of malnutrition with social and economic factors in general and environmental factors in specific such as sanitation facilities and drinking water sources for Pakistan. The authors display a well-organized overview of current and past research findings in the field and identify the gap. The methodology is appropriate for the research. The theoretical framework is sound and well applied. Findings were clearly stated and applied to other current research data in the discussion section. The authors did not include a limitations section.

It is important to keep in mind who the audience of this research study will be. It is heavy on statistical analysis and tables. Some edits for grammar are needed. This is an excellent study addressing a very important topic!!

Abstract: 

The abstract includes all necessary sections.

Introduction:

The section was very organized, and the purpose of the study was rooted in the existing research data regarding malnutrition. It might be beneficial to provide some definition of the terms such as stunting or wasting. This would increase the accessibility of content for a wider audience.

Methods:

The authors clearly outlined all procedural aspects in terms of data analysis.

Findings: 

Statistical analysis appropriate. Tables demonstrated results clearly.

Discussion.

At the beginning of the discussion section, authors compared some of the results from the present study to findings from other research related to the area of study.  The conclusion section is concise and clearly outlines the potential impact of this research and the societal impact of the topic.

Line Feedback:

Line 33: avoid using “we” instead: Authors conclude…

Line 45-46: provide definitions for “stunting”, “wasted”

Linen 48-49; There is a leap here. Also, be aware of person-first language “stunted children”. I am not sure why we are looking at earning potentials at this point in the paper.

Line 54: elaborate “different forms of malnutrition”

Line 72: consider “assesses the impact of”

Line 122-123: That is not the hypothesis stated earlier in the paper. If that is author's hypothesis, it needs to be stated at end of the introduction.

Line 145: “we” should be changed to “Authors”

Line 155-156: edit for grammar

Line 175: “we” should be changed to “Authors”

Line 204: change “our”

Line 216-217: Consider rewording: “study is aligned with …”

Line 219” “our study”- needs to be changed – please edit the manuscript for all other “our” or “we”

Line 231-233: rewrite for clarity

Line 239: word choice “odd of diarrhea”

Line 246: word choice “odd” consider “likelihood” or “incidence of”

Line 266: word choice “female education” maybe consider: education and employment for women

Author Response

Dear Reviwer,

Thanks for your comments. Please see attached for our response. 

best regrads,

Yang Cao

Reviewer 2 Report

No major revisions to perform

Author Response

Dear Reviewer, 

Thanks very much for reviewing our manuscript

Best regards,

Reviewer 3 Report

The data extracted from the Pakistan Demographic and Health Survey 2017-18 has used to used to determine the Socio-economic and environmental determinants of malnutrition in children under the age of 3 years. Although there is a lot of published data on the issue, the present study has tried to present the etiology of malnutrition from a different perspective. The study methods and model used are acceptable and the conclusion are well presented. However, the improved water and improved sanitation categories are not the only one that significantly contribute to the etiology of malnutrition in Pakistan. The authors must have to explain this point in with further sound arguments. The conclusion presented supports my comments and therefore the recommendations made should be reconsidered.  The language of the manuscript is acceptable but requires some editorial corrections. Overall, I recommend the manuscript for its publication with these minor corrections.

Author Response

Dear Reviewer, 

Thanks very much for the comments. Please see attached for your response. 

Best regards,
